# Neural mechanisms for the attention-mediated propagation of conceptual information in the human brain

**David Acunzo\*, Damiano Grignolio, Clayton Hickey**\*

Centre for Human Brain Health and School of Psychology, University of Birmingham, Birmingham, United Kingdom.

\* GDK3UB@uvahealth.org (DA); c.m.hickey@bham.ac.uk (CH)

## Abstract

The visual environment is complicated, and humans and other animals accordingly prioritize some sources of information over others through the deployment of spatial attention. Cognitive theories propose that one core purpose of this is to gather information that can be used in downstream cognitive processes, including the development of concepts and categories. However, neuroscientific investigation has focused closely on the identification of the systems and algorithms that support attentional control or that instantiate the effect of attention on sensation and perception. Much less is known about how attention impacts the acquisition and activation of concepts. Here, we use machine learning of EEG and concurrently recorded EEG/MRI to temporally and anatomically characterize the neural network that abstracts from attended perceptual information to activate and construct semantic and conceptual representations. We find that variance in the amplitude of N2pc—an event-related potential (ERP) component closely linked to selective attention—predicts the emergence of conceptual information in a network including prefrontal, posterior parietal, and anterior insular cortex. This network appears to play a key role in the attention-mediated translation of perceptual information to concepts, semantics, and action plans.

## Introduction

The visual environment contains more information than we can evaluate or act upon. To behave adaptively, we prioritize some information sources over others using a set of mechanisms collectively referred to as *selective attention* [1,2]. Early work characterized attention in now-famous metaphors, suggesting a spotlight or filter that acts in retinotopic space to enhance sensation and perception [3,4]. Guided by these ideas, neuroscientific investigation has focused closely on identification of the systems and algorithms that impact sensory cortex [1,5,6] and on characterization of the brain networks that control these mechanisms [7–9]. The emergent model suggests that strategic selection begins with the derivation of schemas and templates from conceptual and semantic knowledge. These are stored in working memory at frontal brain sites and translated through the frontal eye fields and posterior parietal cortex into perceptual hypotheses and biases of sensory activity [1,10].

**Data availability statement:** All data is available from the University of Birmingham UBIRA eData repository at edata.bham.ac.uk/1236/

**Funding:** All authors are supported by the European Research Council under the European Union Horizon 2020 Research and Innovation Program (Grant Agreement 804360 to CH). The funder had no role in study design, data collection and analysis, decision to publish, or preparation of the manuscript. https://research-and-innovation.ec.europa.eu/funding/funding-opportunities/funding-programmes-and-open-calls/horizon-2020_en

**Competing interests:** The authors have no competing interests to declare.

**Abbreviations:** BG, basal ganglia; CB, cardioballistic; dmSFG, dorsomedial superior frontal gyrus; ECG, electrocardiogram; EPI, echo planar imaging; ERP, event-related potential; FOV, field-of-view; GLM, general linear model; ICA, independent component analysis; IFG, inferior frontal gyrus; IPL, inferior parietal lobules; LDA, linear discriminant analysis; MFG, middle frontal gyrus; OFC, orbitofrontal cortex; PC, posterior cingulate; ROIs, regions of interest; SPL, superior parietal lobules; TE, echo time; TPJ, temporal-parietal junction; TR, repetition time; VCG, vectorcardiogram; LOC, lateral occipital cortex

Research on the neurophysiology of attention has thus focused closely on how concepts and categories are used to guide sensation and perception. In contrast, our understanding of the reverse inference—of how attention supports the activation and construction of concepts—is relatively poor. That is, we know that attention plays a key role in information gathering and the generation of high-level knowledge [11,12]. And, while there is debate about the degree to which concepts are grounded in sensorimotor systems [13], we know that the semantic representations that emerge in frontal and temporal cortex abstract from perception [14–17]. Theoretical treatments of attention propose that a core purpose of selection is to activate and construct these conceptual representations, in this way supporting cognition and decision-making [11,12], but we know little of how this unfolds in the brain.

To address this, in the current paper we identify and characterize neural representations of abstracted information that vary as a function of the quality of attentional selection. This approach has been adopted in earlier work. For example, results show that the semantic category of a visual stimulus is encoded in patterns of fMRI activity in the ventral and dorsal visual cortex [18,19] and that these representations vary as a function of their behavioral relevance, consistent with an effect of attention [20–22]. However, this pattern has largely been identified in brain areas that are likely to encode visual features, making it unclear that it reflects an effect on category information rather than on perception of the visual features that happen to be shared across category examples [23–25]. To circumvent this issue, a separate research line has employed multi-modal stimuli that activate the same concept—for example, by presenting an image of an apple and the word 'apple'. This has identified category information in areas that do not carry perceptual signals, such as anterior temporal and prefrontal cortex [15–17,26]. However, the role of attention in the activation or construction of these representations remains unclear.

As a result, there are holes in our understanding of the neural mechanisms that support information gathering and abstraction. Theory suggests that that information selected through the deployment of attention activates or constructs neural representations of concepts and categories [11,12], but this has not been verified, and we do not know where in the brain this takes place. Are all such high-level representations sensitive to attention, or is the effect of attention limited to discrete brain networks? And what does this mean for our understanding of the functionality of these areas? Here, we use novel methodology and analyses to identify the effect of attention on information about the semantic category of visual stimuli. Our approach relies on identification of covariance between a well-characterized EEG index of spatial attention—the N2pc [27]—and an independent measure of semantic category information derived using machine learning of neuroimaging data. Experiment 1 employs high-density EEG in the identification of this information to allow for precise temporal resolution. Experiment 2 employs concurrently recorded EEG and MRI to provide precise spatial localization.

## Results

Participants in both experiments (*n* = 44, *n* = 31) were asked to complete the same task without moving their eyes. This involved reporting, via button press, the category of a target object (Fig 1A) that appeared at a cued location (Fig 1B), while ignoring a set of non-targets that included visually degraded objects and a single recognizable distractor. Objects were selected from a set of images that were visually heterogeneous within each category. Participants completed the task quickly (E1: 1,076 ms, 185 SD; E2: 1,088 ms, 240 SD) and accurately (E1: 87.6%, 10.6 SD; E2: 89.9%, 7.0 SD).

In both experiments, we measured the quality of attentional processing using the N2pc component of the visual event-related potential (ERP) [29]. The N2pc emerges over the

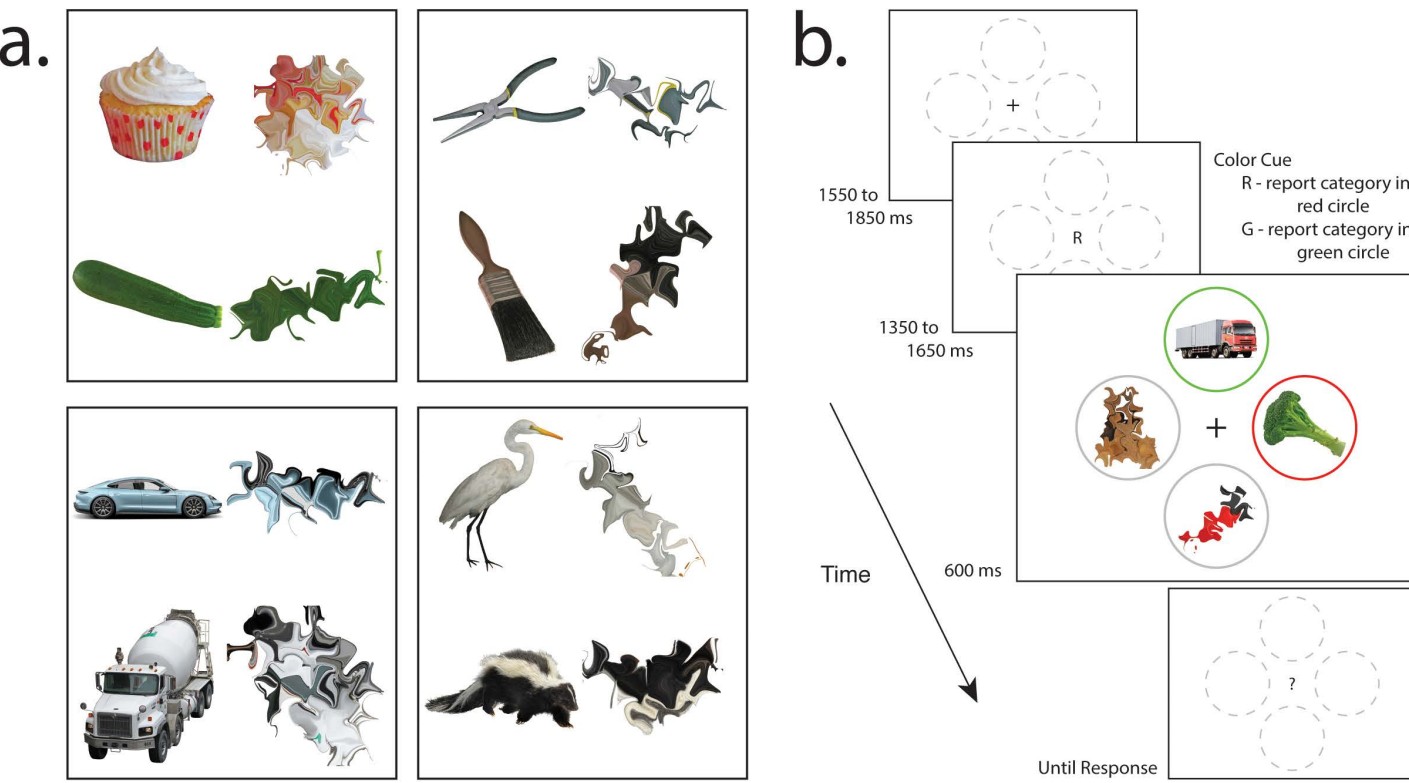

**Fig 1. Experimental task.** (**a**) Complete, recognizable examples of object categories and morphed, unrecognizable versions of the same images. In Experiment 1, the stimuli set was composed of 40 complete images in each of the four categories. In Experiment 2, a subset of 30 complete images per category was employed. All images presented in this figure are included in the BOSS image database [28] and written permission to employ these images in this figure has been kindly granted by the owner. (**b**) Experimental procedure. The target was cued by a central letter, either "R" or "G" to denote red or green. Participants had to report which category appeared in the cued circle via right-hand button response. Two categories were associated with index finger response and two with middle finger response, with response mapping counterbalanced across participants.

posterior cortex contralateral to the location of an attended object [29,30]. It increases in amplitude as circumstances require a greater investment of attentional resources [31,32] and is therefore used as an index of the depth of attentional processing [29,33]. In Experiment 2, we conduct additional exploratory analyses of how lateral ERP effect preceding or following N2pc also impacted the emergence of category information.

We tracked the emergence of category information in the brain using linear discriminant analysis (LDA) to classify neuroimaging data. We trained models to classify the object category of the target or distractor based on normalized patterns of evoked brain activity, subsequently testing these models on retained data to calculate model classification accuracy.

## Measuring attentional selection and category information in EEG

In each experimental trial, the target object could appear to the left or right of a central fixation mark or directly above or below it; when the target was presented laterally, the single recognizable distractor was presented on the vertical, and vice versa (Fig 1B). This design was adopted as stimuli on the vertical meridian of the visual field initially create activity of equal magnitude in both cortical hemispheres, and therefore attentional selection of these stimuli does not generate the lateralized EEG voltage underlying N2pc [34]. By presenting stimuli on the vertical we were able to isolate the N2pc elicited exclusively by the target or distractor [30,35].

The N2pc emerged prominently in both experiments (Fig 2A and 2D) and was larger in amplitude for targets than recognizable distractors (Fig 2B and 2E). It was preceded by an earlier, positive-polarity lateral voltage that reflects an imbalance in sensory activity between the visual hemispheres. In our stimulus displays, there is a difference in the complexity and spatial frequency of the complete object, presented on one side of the screen, and the unrecognizable degraded object, presented on the other (Fig 1B). This kind of imbalance creates short-latency

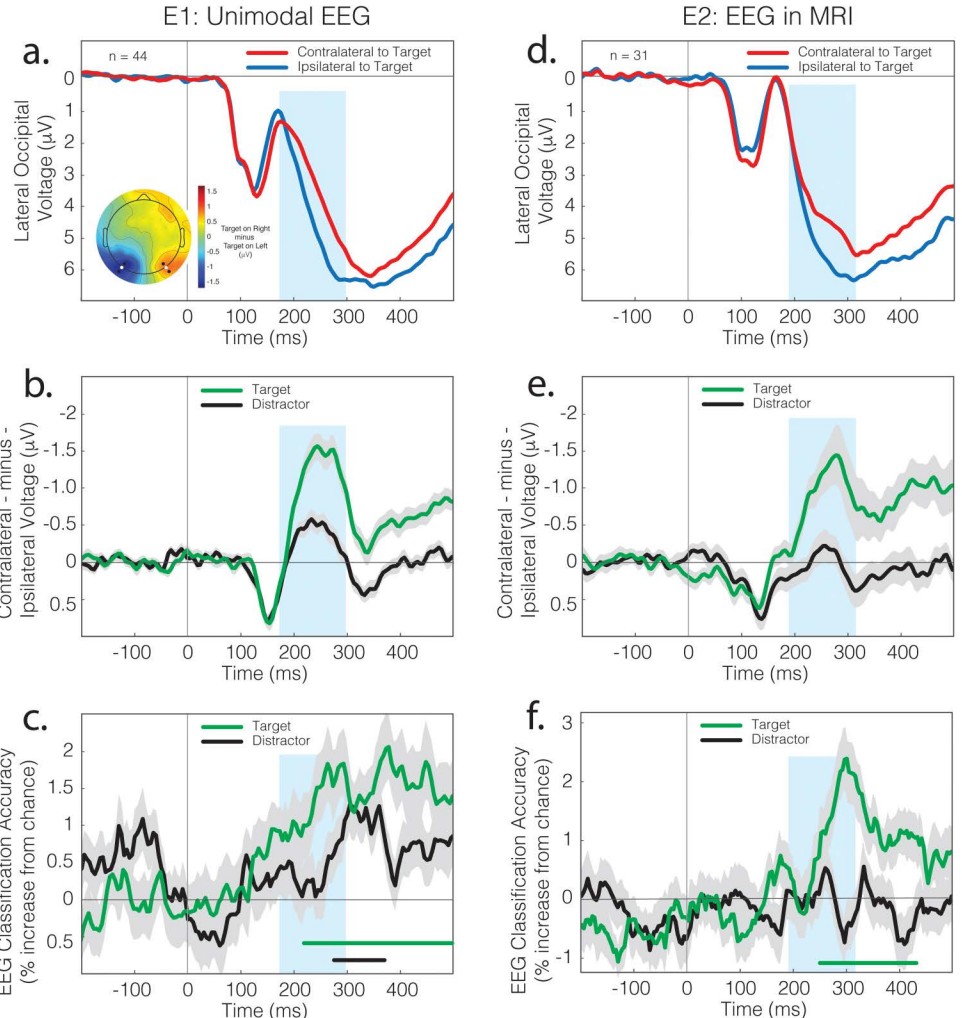

**Fig 2. Results from EEG.** (**a**) ERP results from Experiment 1. The N2pc is evident as a negativity in the posterior waveform recorded over the cortex contralateral to an attended stimulus. The blue box identifies a 125 ms interval centered on the cross-conditional peak of the N2pc. The topographical plot reflects the voltage difference between conditions when the target was in the left vs. right visual field; the N2pc expresses as a negativity in the left hemisphere but a positivity in the right hemisphere due to the subtraction. Contralateral and ipsilateral ERPs reflect mean signal across the 2 sets of 3 lateral and posterior electrodes identified in the topography by a larger marker. (**b**) Contralateral-minus-ipsilateral difference waves from Experiment 1. The N2pc is evident as negative deflection beginning around 175 ms post-stimulus. Shading here and in other panels indicates bootstrapped SEM. (**c**) Time-resolved classification of the target and distractor object category in Experiment 1. As there were four classes, the chance performance was 25%. (**d**) ERP results from Experiment 2. These reflect signal at the lateral posterior electrodes identified in white in the topography presented in panel A. (**e**) Contralateral-minus-ipsilateral difference waves from Experiment 2. (**f**) Time-resolved classification of target and distractor object category in Experiment 2.

laterality in the visual ERP (i.e., PPN) [36]. The insensitivity of this effect to the behavioral relevance of the evoking stimulus distinguishes it from the subsequent N2pc.

Target and distractor category information was identified in the EEG using classification analysis. For each sample in the epoched data, an LDA model was trained on results observed in a ~85 ms window centered on that observation. A cross-fold validation scheme—described further below—was employed to test the model in each of these windowed samples, generating a time-course of classification accuracy. As illustrated in Fig 2C, when the target appeared laterally, the target category could be reliably classified from approximately 215 ms post-stimulus in Experiment 1 (Fig 2C) and from 240 ms in Experiment 2 (Fig 2F). Distractor classification also emerged in Experiment 1, though later and with reduced magnitude and extent.

Importantly, we designed the experiments such that the classification of object category could not leverage the EEG variance underpinning N2pc. In all classification analyses, model creation and validation were based on sets of trials that contain the same numbers of relevant vehicles, tools, foods, and animals, and roughly the same number of left- and right-lateralized stimuli within each of these categories. The location of the object—and thus the laterality of the N2pc in left or right visual cortex—provided no information about its category.

## Coupling variance in EEG indices of attentional selection to variance in EEG-derived category information

We leveraged natural variance in the quality of attentional selection to identify covariance in N2pc and category classification accuracy. This variance stems in part from changes in participant vigilance, task engagement, and fatigue, but also reflects confusion and the misallocation of attention to distractor objects. We expected that when the quality of attentional deployment to the target was high, this would cause the emergence of information about the attended object in the neural signal.

To test this, we iteratively resampled the data in a bootstrap routine, building classification models and measuring the N2pc in each instance (Fig 3A). For each participant, target-lateral and distractor-lateral trials were separated into 20 chunks, each containing an equal number of trials where the classified object was a food, vehicle, animal, or tool. The chunks were allocated repeatedly into two subsets. The larger set—constituting 17 chunks or 85% of trials—was used to train a classification model that identified the target category. The smaller set—constituting the remaining three chunks or 15% of trials—was used both to test the model and to measure the target- or distractor-elicited N2pc.

For each of the target-lateral and distractor-lateral conditions, we iterated this process of sampling and model building 1,140 times, reflecting all possible combinations of 3 chunks taken from 20. The classification accuracy illustrated in Fig 2C and 2F reflects the mean of these values across iterations and subsequently participants. In each iteration, we measured the N2pc as mean voltage in a 125 ms interval centered at the peak (as identified from data collapsed across iterations, conditions, and participants; 173–298 ms). We measured mean classification accuracy in this same interval, resulting in paired sets of 1,140 N2pc and classification accuracy values for each analysis and participant. As illustrated in Fig 4A, within-participant results commonly showed a relationship between target-elicited N2pc and target classification accuracy, with negative-voltage N2pc amplitude predicting an improvement in target classification. The mean relationship is plotted in Fig 4B alongside results from each of the 44 participants.

One interpretation of this pattern is that robust attentional selection of the target—reflected in larger target-elicited N2pc—caused the EEG to carry more information about the target category. However, an alternative is that both N2pc and target information covaried

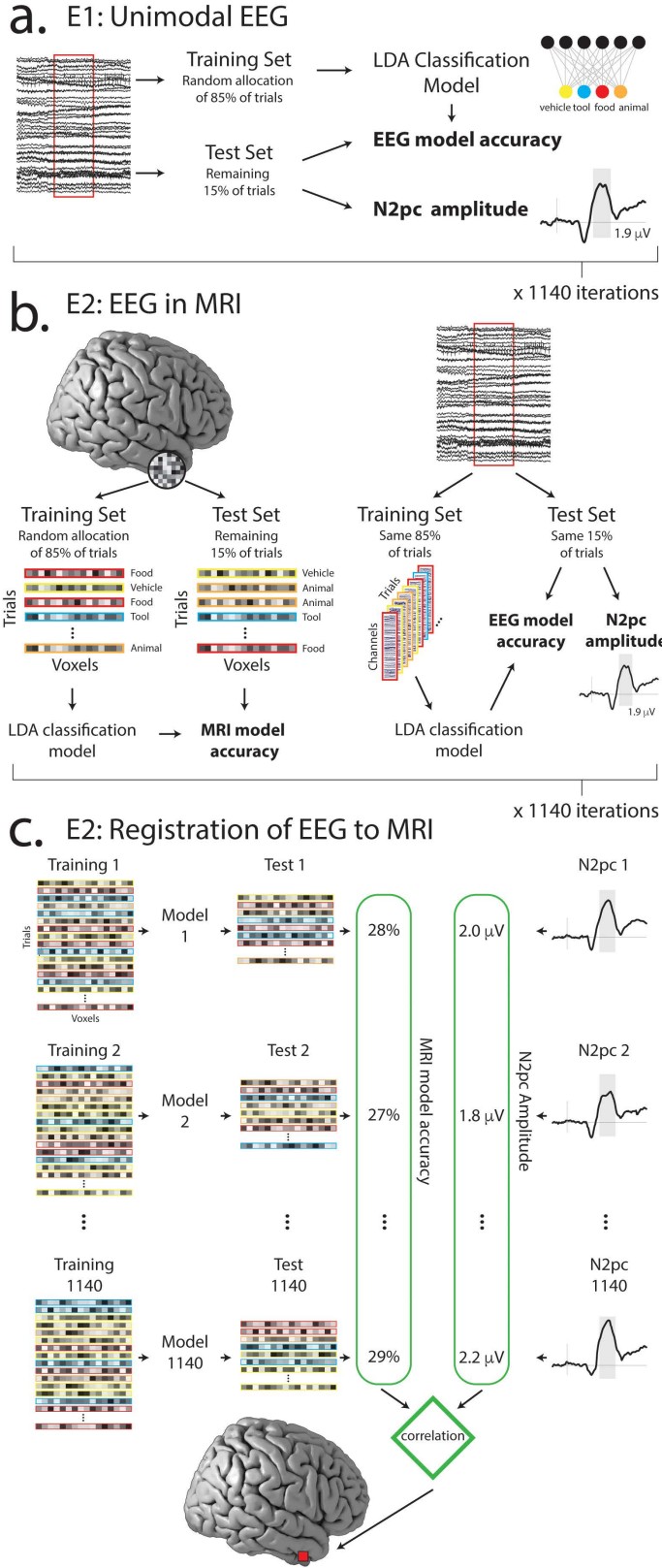

**Fig 3.** **(a) Analysis of EEG data in Experiment 1.** Trials were repeatedly partitioned into training and test sets. Classification accuracy and mean N2pc amplitude were measured for each test set, resulting in two dependent measures

(identified in bold). (**b**) Analysis of MRI and EEG data in Experiment 2. The imaging modalities were linked during trial partitioning, such that the same trials appeared in both MRI and EEG training and test sets. This resulted in three dependent measures (identified in bold). (**c**) To link EEG and MRI results in Experiment 2, we correlated MRI classification accuracy with N2pc amplitude observed across the set of 1,140 sampling iterations. This occurred for each searchlight sphere.

as a function of participant arousal or task engagement without any more direct relationship. We conducted a second analysis to probe this alternative. Here, we related target information to distractor-elicited rather than target-elicited N2pc. If the N2pc and category information broadly covary, this analysis should identify the same relationship as described above. However, as illustrated in Fig 4C, the magnitude of distractor-elicited N2pc predicted a decrease in target classification accuracy.

We used hierarchical linear modeling of EEG and model assessment to statistically evaluate these patterns [37]. An initial model of classification accuracy included a single random effect for the per-participant intercept. This was improved by adding a fixed effect—*condition*—that identified if the target or distractor was being classified (alongside a corresponding random effect for the per-participant slope; Akaike information criterion (AIC) −8746) [38]. The model was further improved by adding a fixed effect for *N2pc* amplitude (and corresponding random effect; AIC −702) and the interaction of these factors (and corresponding random effect: AIC −896).

$$\text{classification\_accuracy} \sim \text{N2pc} * \text{condition} + \left( \text{N2pc} * \text{condition} | \text{participant} \right) \quad (1)$$

ANOVA analysis of this model identified an interaction of *condition* and *N2pc* ($F(1, 42.6) = 13.72$, $p < 0.001$), reflecting a significant difference in the slope of the linear fits illustrated in Fig 4B and 4C. Follow-up modeling examined the relationship between N2pc and category information separately for each of the conditions.

$$\text{classification\_accuracy} \sim \text{N2pc} + \left( \text{N2pc} | \text{participant} \right) \quad (2)$$

Target-elicited N2pc reliably predicted improved target classification ($F(1, 41.6) = 5.366$, $p = 0.026$), but distractor-elicited N2pc reliably predicted degraded target classification ($F(1, 42.0) = 6.130$, $p = 0.017$). No pattern was identified in a separate analysis of distractor-elicited N2pc and distractor classification accuracy (Fs < 1).

Results from Experiment 1 demonstrate that the quality of attentional selection of the target, reflected in target-elicited N2pc, predicts an increase in the target category information carried by EEG in the same 125 ms time period. In contrast, misallocation of attention to the distractor, reflected in distractor-elicited N2pc, predicted degraded target category information. While causation cannot be inferred from this correlational result, the pattern is in line with the idea that where better attentional selection of the target drives the emergence of target category information, attentional selection of the distractor creates a degraded representation of the target.

## Measuring category information in unimodal fMRI

In Experiment 2, participants completed the same task while EEG and fMRI were concurrently recorded (Fig 3B). We began analysis with the identification of category information in the unimodal fMRI signal. This analysis does not consider the laterality of the target or distractor image. We estimated single-trial fMRI responses using the GLMsingle toolbox [39] and identified category information in brain space using spatial searchlights and LDA classification. In this analytic

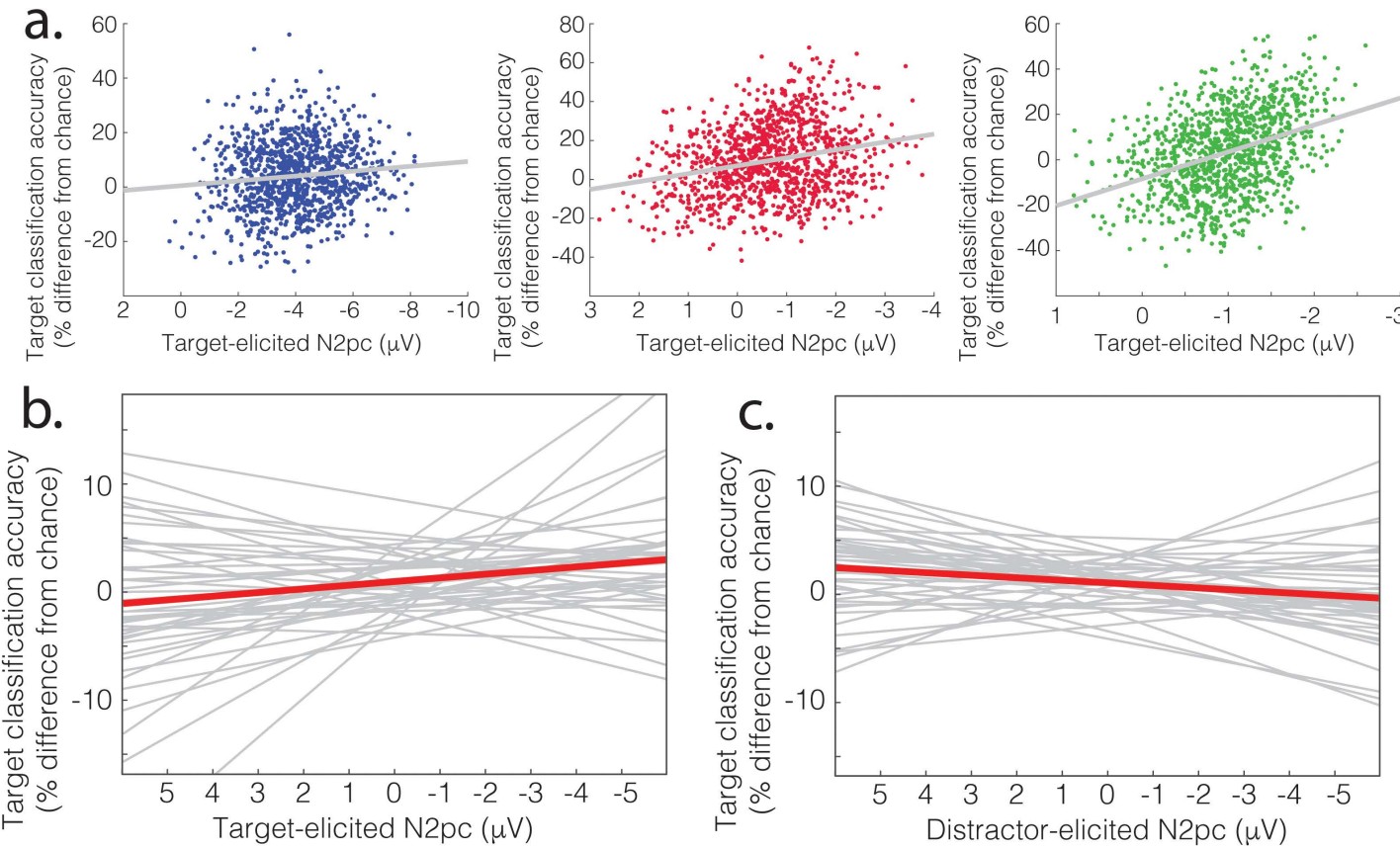

**Fig 4. Linking N2pc and EEG category information in Experiment 1.** (**a**) Within-participant results from three example participants. Across the 1,140 sampling iterations, the target-elicited N2pc observed in each sample predicts target classification accuracy in that sample. (**b**) Least-square linear fit of target-elicited N2pc and target classification accuracy for each participant, with mean relationship identified in red. (**c**) Linear fit of distractor-elicited N2pc and target classification accuracy for each participant, with mean relationship identified in red.

approach, a sphere is defined around each voxel, and classification is conducted based on the pattern of activity observed in the subset of voxels within this volume [40]. Searchlight spheres had a diameter of 7 voxels (14 mm) and classifiers were trained and tested using 4-fold cross-validation. This identified an overlapping set of brain areas that contained information about both the target and the distractor, alongside a more limited set of areas that exclusively contained information about one of these two objects (Fig 5A). Brain areas expressing category information for both target and distractor included ventral visual cortex; inferior and superior parietal lobules (IPL/SPL); and pre-motor, motor, and somatosensory cortex. Target information also emerged in the left inferior frontal gyrus (IFG), bilateral orbitofrontal cortex (OFC), and precuneus/cingulate cortex; distractor information extended from the IPL into the left temporal-parietal junction (TPJ). Statistical analysis identified that the bilateral lateral occipital cortex (LOC), IPL/SPL, left IFG, left OFC, posterior cingulate (PC), and left basal ganglia (BG) all reliably contained more information about the target than the distractor (Fig 5B). No area reliably contained more information about the distractor than the target.

## Coupling variance in EEG indices of attentional selection to variance in MRI-derived category information

In Experiment 1, we used a resampling approach to identify within-participant covariance in independent measures derived from EEG. In Experiment 2, we used a similar approach to link

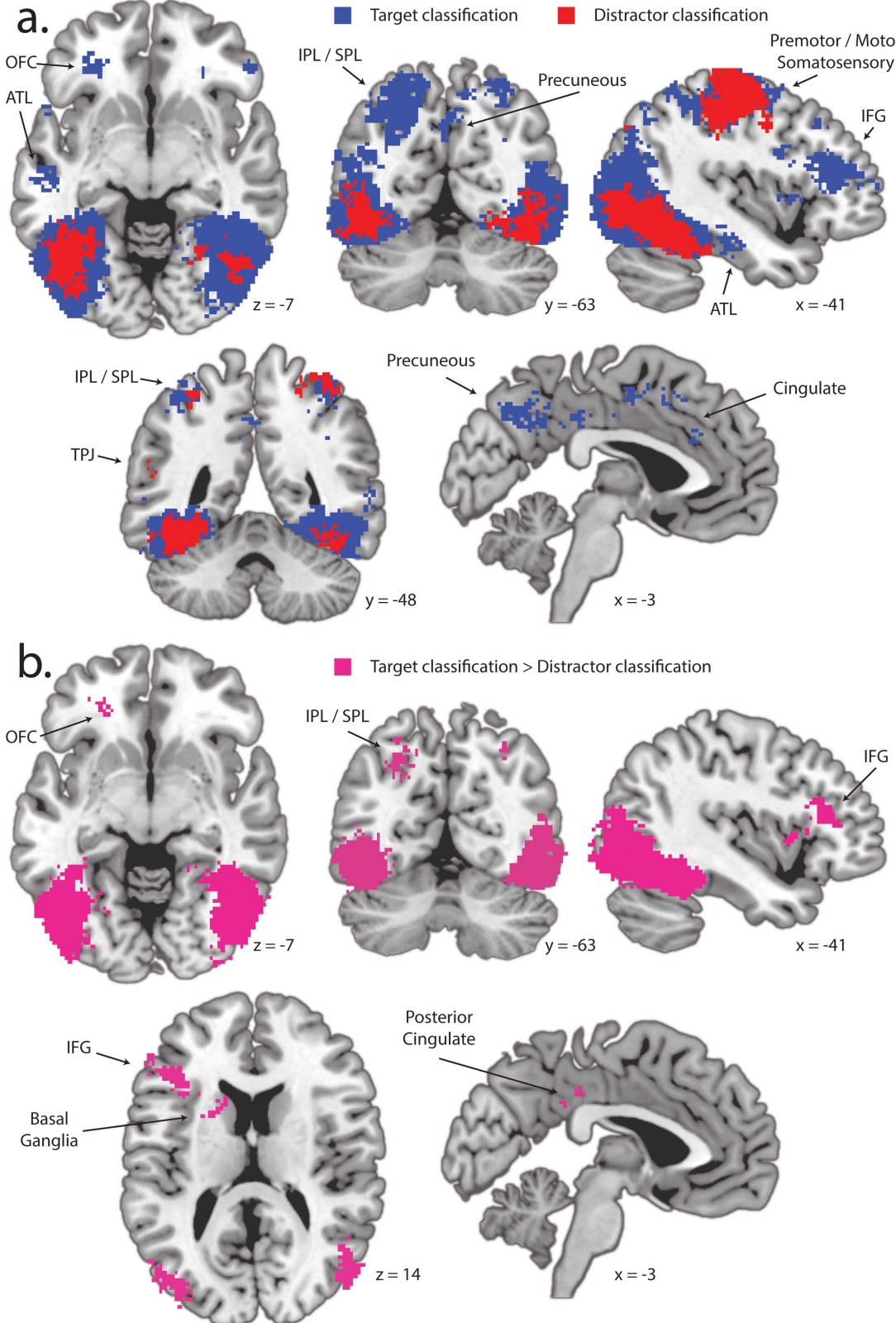

**Fig 5. Results from MRI classification.** (**a**) Searchlight analysis identifying the location of target and distractor information. Target and distractor category information emerge with particular strength in the aspects of occipital and posterior temporal

lobes associated with the ventral visual stream. (**b**) Voxels identified here reflect clusters expressing more information about the target than about the distractor. In both panels, the largest clusters emerge in the ventral visual cortex; other cluster locations are identified. OFC, orbitofrontal cortex; IPL, inferior parietal lobe; SPL, superior parietal lobe; IFG, inferior frontal gyrus; TPJ, temporal-parietal junction. The left hemisphere is represented on the left here and in subsequent figures.

variance in measures derived from the two imaging modalities (Fig 2C). For each participant, trials were again separated into 20 chunks of equal size, with 17 chunks assigned to the training set, and 3 chunks to the test set. This process of sampling and model building was iterated 1,140 times for each of the ~$2.1 \times 10^5$ searchlight spheres. In each of these iterations, the N2pc was measured in the test set as the mean voltage across a 125 ms interval centered at the peak (as identified from data collapsed across iterations, conditions, and participants; 189–315 ms). This generated a paired set of 1,140 N2pc and classification accuracy values for each searchlight sphere. We calculated the Fisher-transformed Pearson correlation for each of these sets, inserting the resulting values into a new brain volume at the location of each searchlight center. As illustrated in Fig 6, this identified three clusters where target-elicited N2pc amplitude predicted MRI-derived target category information. This included a cluster in the left IPL/angular gyrus extending into the SPL (92 voxels; Fig 6A), a cluster that spanned the putamen, globus pallidus, insula, and anterior temporal lobe in the right hemisphere (84 voxels, Fig 6B), and a cluster that spanned the bilateral ventromedial prefrontal cortex and OFC (VMPFC/OFC; 163 voxels, Fig 6C).

While these correlational results cannot demonstrate causation, one interpretation here is that attentional selection of the target—reflected in target-elicited N2pc over the occipital cortex—caused the emergence of conceptual information in the identified brain areas. However, as in Experiment 1, the alternative is that N2pc and category information broadly covaried as a function of other variables associated with participant state, like arousal. We again conducted a control analysis to test this, relating target category information to distractor-elicited rather than target-elicited N2pc. We created three regions of interest (ROIs) that described the clusters illustrated in Fig 6, calculated classification accuracy using the entire set of voxels within each of these ROIs, and used our 1,140-fold technique to relate this information to the N2pc. This showed that the relationship between target category information and target-elicited N2pc was significantly greater than the relationship between target category information and distractor-elicited N2pc in all three ROIs (IPL/SPL: $p = 0.005$; insula: $p = 0.027$; VMPFC/OFC: $p < 0.001$). Distractor-elicited N2pc predicted a reduction in MRI-derived target category information in the VMPFC/OFC, much as emerged in the analysis of unimodal EEG results from Experiment 1 ($p = 0.019$; other areas ps > 0.230).

We conducted whole-brain analyses to relate distractor-elicited N2pc to variance in target or distractor information but failed to identify any set of voxels where these relationships survived statistical correction. Analysis of distractor-elicited N2pc and distractor information based on ROIs defined from results illustrated in Fig 6 also failed to identify significant results. The absence of reliable effects involving distractor information may in part reflect the limited magnitude and scope of this information in the MRI data (Fig 5).

## Coupling variance in EEG indices of attentional selection to variance in fMRI-derived category information: Time-course

To this point, we have focused on the N2pc measured at a fixed latency. However, laterality in the EEG emerges both before and after the N2pc; as discussed above, early laterality emerges

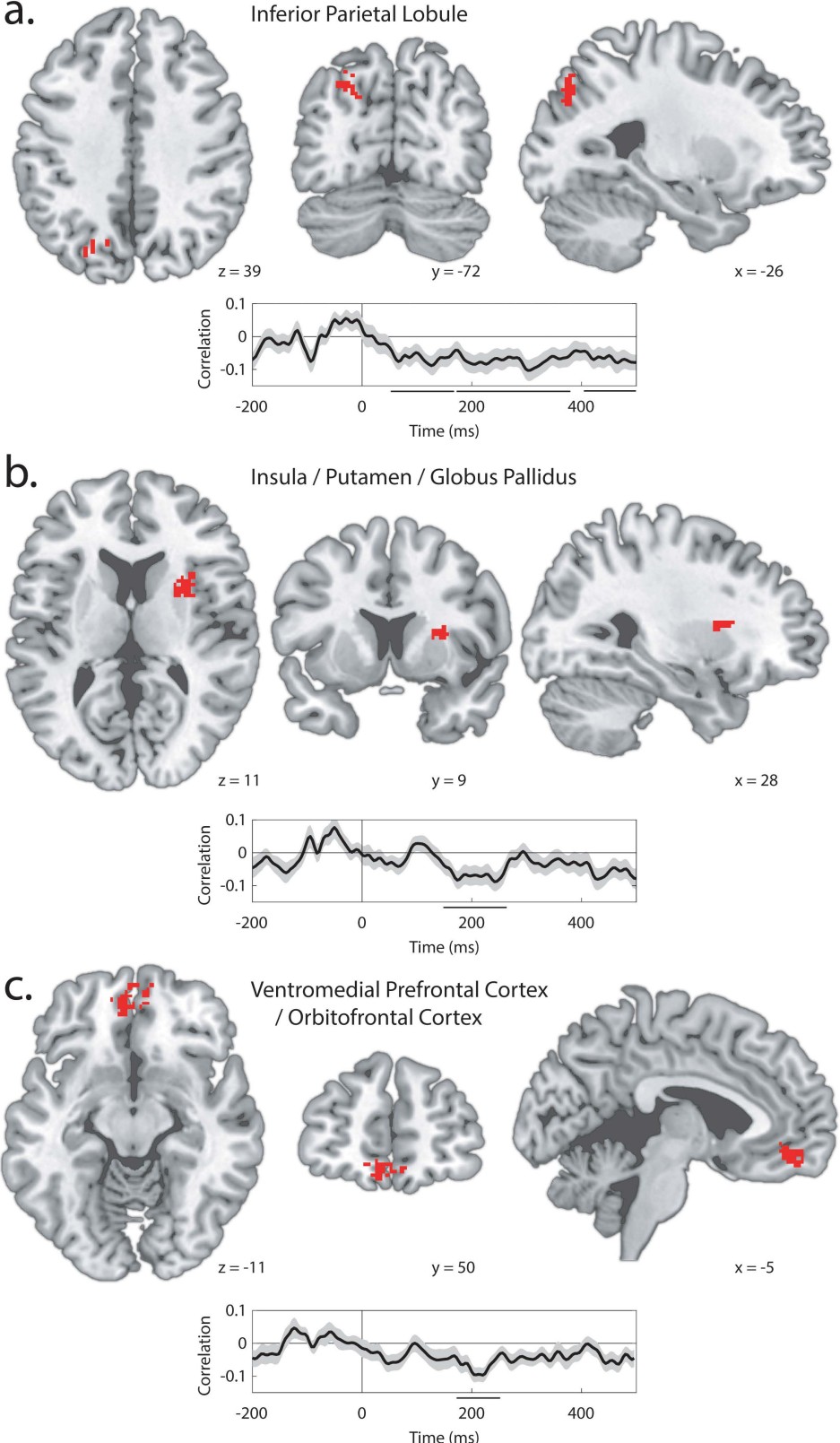

**Fig 6. Linking N2pc and MRI category information in Experiment 2.** Results illustrate brain regions where target category information is predicted by target-elicited N2pc amplitude. Time-course insets show the time-resolved

correlation for each MRI cluster. Significant time clusters are indicated by black lines below the *x*-axis and shading reflects bootstrapped SEM. Location and time-course for (**a**) the Inferior Parietal Lobule cluster, (**b**) the Insula/ Putamen/Globus Pallidus cluster, and (**c**) the Ventromedial Prefrontal Cortex/ Orbitofrontal Cortex cluster.

as a function of sensory activity, but also as a product of stimulus novelty [41] and distractor suppression [41–43]. Later, laterality has been closely linked to the encoding and maintenance of stimuli in visual working memory [44]. We expanded our analysis to explore the role of pre- and post-N2pc EEG laterality in the control of conceptual encoding in the brain. For each of the ROIs generated from results illustrated in Fig 6, we related variance in each lateral EEG sample to variance in MRI-derived target category information. This generated a time-course of the relationship between lateral occipital EEG- and MRI-derived target category information for each ROI. These are illustrated in Fig 6. In the IPL, the relationship between EEG laterality and target category information emerged very quickly and sustained (Fig 6A). The pre-N2pc relationship may reflect distractor suppression that guides selective processing of the complete object rather than the degraded object in the contralateral visual hemifield [43,44]. The post-N2pc relationship, in contrast, is likely to index the maintenance of visual working memory representations of the target in the posterior parietal cortex [45,46]. We expect that these mechanisms would operate in concert with the N2pc to guide selective processing and information gathering. In contrast, in the VMPFC and insula clusters, the relationship between EEG laterality and MRI-derived target information is relatively discrete to the latency interval of the N2pc (Fig 6B and 6C). These results suggest that the specific selective processes indexed in N2pc are closely linked to the propagation of target category information to these areas.

## Coupling variance in EEG indices of attentional selection to variance in fMRI voxel activity

In the text above, we have identified a relationship between N2pc amplitude and the emergence of MRI-derived category information. This analysis was designed to identify the effect of N2pc on the propagation of category information derived from visual stimuli. However, the data also offer an opportunity to identify how N2pc is predicted by univariate voxel activity.

Our analysis of the relationship between univariate fMRI and N2pc was conceptually similar to our analysis of multivariate fMRI and N2pc. Chunked per-participant results were exhaustively partitioned into subsets of 3 and 17. In each of the resulting 1,140 test sets, the N2pc was measured in a 125 ms window around its peak and mean activity was calculated for each of the $\sim 2.1 \times 10^5$ voxels in cortex and subcortex. This generated paired sets of 1,140 N2pc and activation values for each voxel. For each participant, we calculated the Fisher-transformed Pearson correlation for each of these paired sets, inserting the values into a new brain volume at each voxel location.

As illustrated in Fig 7A, this identified three voxel clusters where variance in target-elicited N2pc amplitude predicted variance in voxel activation: the middle frontal gyrus (MFG; 377 voxels), the IPL extending into the TPJ (486 voxels), and the dorsomedial superior frontal gyrus (dmSFG; 296 voxels). These areas are largely independent of those identified in the corresponding analysis of category information. Significant results emerged in the IPL in both analyses, but the clusters did not overlap, with N2pc predicting activation in voxels anterior and lateral to those identified in the analysis of category information.

As in the analysis of category information, we examined the relationship between distractor-elicited N2pc and voxel activity as a control. Whole-brain analysis did not identify any brain

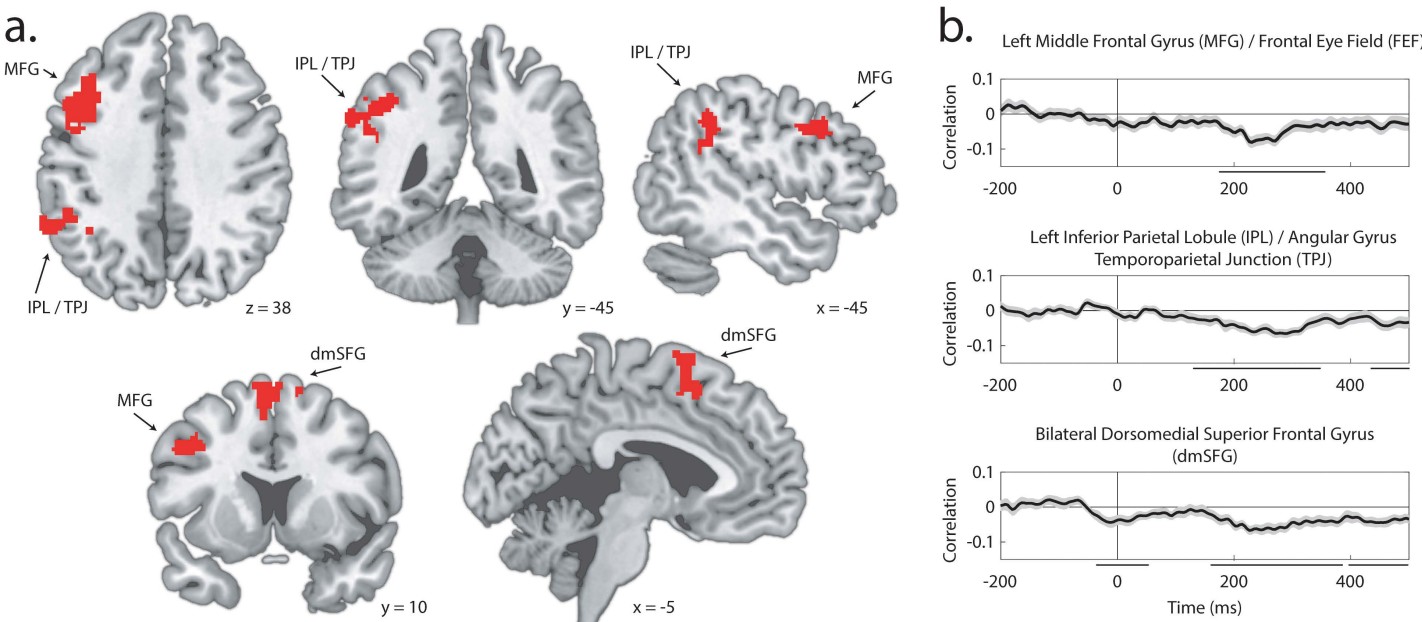

**Fig 7. Linking N2pc and MRI voxel activation in Experiment 2.** Results illustrate brain regions where target-elicited N2pc amplitude is predicted by voxel activity. (**a**) Location of the three identified clusters: Left Inferior Parietal Lobule/Angular Gyrus/Temporoparietal Junction, Left Middle Frontal Gyrus/Frontal Eye Field, and Bilateral Dorsomedial Superior Frontal Gyrus. (**b**) Time-resolved correlation for each MRI cluster. Significant time clusters are indicated by black lines below the *x*-axis and shading reflects bootstrapped SEM.

area where the distractor-elicited N2pc predicted voxel activity after statistical correction. When we defined ROIs based on the results from whole-brain analysis illustrated in Fig 7, there was no significant relationship between distractor-elicited N2pc and mean voxel activity in these areas, and the relationship between target-elicited N2pc and voxel activity was significantly greater than the relationship between distractor-elicited N2pc and voxel activity (IPL: $p$ = 0.007; dmSFG: $p < 0.001$; MFG: $p$ = 0.007).

## Coupling variance in EEG indices of attentional selection to variance in fMRI voxel activity: Time-course

As in the analysis of category information, we expanded our analysis to test the power of pre- and post-N2pc EEG laterality to predict univariate voxel activity. As illustrated in Fig 7B, a relationship between lateral EEG and voxel activity emerges sooner in IPL than in MFG, though both effects are in the 175–350 ms interval where the N2pc can emerge. Both left IPL and left MFG have been closely linked to the extraction of semantic content during reading [47,48], and left and right IPL and MFG have roles in the orientation and reorientation of spatial attention [8,49]. One interpretation of this finding is that the emergence of IPL and MFG reflect control in the deployment of attention for the extraction of semantic information.

The relationship between lateral EEG and activity in the dmSFG, in contrast, emerges from very early, including latencies that precede stimulus onset. Among other functions, this portion of medial frontal lobe contains eye fields and has been associated with fixation, including during smooth pursuit eye movements [50]. In the current results, trials with eye movements substantively preceding stimulus onset were not excluded from analysis; the association of activity in the dmSFG with EEG laterality underlying the N2pc may reflect oculomotor control supporting orientation of the eyes to the fixation mark or in maintenance of the eyes

at that location. Similarly, later association of dmSFG with EEG laterality in the N2pc and post-N2pc interval may reflect the involvement of this brain area in the inhibition of saccadic movements to covertly attended locations.

## Discussion

We leveraged natural variability in the quality of attentional selection to identify the temporal and anatomic characteristics of the attention-mediated propagation of conceptual information in the human brain. In Experiment 1, we find that the amplitude of target-elicited N2pc, an index of spatial attention, predicts that the evoked EEG signal will carry more information about the semantic category of an attended object (Fig 4B). This relationship emerges quickly, with attention-mediated category information emerging in the same 173–298 ms window used in measurement of the N2pc. The misallocation of attention to task-irrelevant stimuli, reflected in distractor-elicited N2pc, predicts a reduction in the quality of target category information in the same interval (Fig 4C). Though these results are correlational, making it impossible to draw firm conclusions regarding causation, the pattern is in line with the idea that attentional selection of the target results in the emergence of category information in brain activity.

Experiment 1 leaves unclear the degree to which the observed relationship between N2pc and category information reflects low-level perceptual encoding. That is, our machine learning approach to the derivation of category information could rely on the representation of low- and mid-level visual features that are shared by examples of a given category, and this is not entirely addressed by our use of a visually heterogenous stimuli set. For example, attentional selection of an image of a vehicle may cause representation of the surface characteristic 'metallic' more often than is the case for any other image category in the experiments. This would provide opportunity for the classifier to leverage the emergence of brain activity associated with the perception of this visual feature to probabilistically infer target category. Stronger attentional accentuation of this brain activity could in this way improve category classification.

One goal of Experiment 2 was therefore to clarify the contribution of visual perception in the relationship between attention and category information. Initial analysis of unimodal fMRI identified a network of brain areas where category information was greater for targets than for distractors (Fig 4B). This prominently included bilateral ventral visual cortex, alongside smaller clusters in bilateral IPL/SPL, left IFG (IFG), left OFC, left BG, and bilateral PC gyrus. These results therefore include aspects of the occipital and posterior parietal cortex that are strongly associated with the representation of visual features and translation of visual information to motor control [51,52], alongside BG structures that may have a role in visual perception [53]. However, also evident is a set of left-lateralized areas including IFG and VMPFC/OFC, alongside bilateral PC, that are not putatively visual in nature, but have been implicated in earlier work as containing abstract semantic representations [26].

Further analysis identified three brain areas where trial-wise variance in target-elicited N2pc amplitude predicted the quality of target category information: the left IPL/SPL; an insula-centered cluster in the right hemisphere including portions of the putamen and globus pallidus; and a bilateral VMPFC-centered cluster extending into medial OFC (Fig 5). These are all nodes in the "semantic network" identified from a large-scale meta-analysis of fMRI studies of semantic representation [26]. The portion of IPL/SPL identified in this analysis was physically close to the posterior parietal cluster identified in the analysis of unimodal fMRI, as detailed above, but was slightly posterior with no overlap, emerging primarily in the angular gyrus. Category information in this area expressed a relationship with ERP laterality over much of 500 ms following stimulus onset, including time periods before and after N2pc.

Activity in this portion of the parietal lobe is implicated in the control of visual attention [54,55] but also in the retrieval and representation of episodic [56–58] and semantic information [49]. In line with other recent theory [49], our tentative account is that this aspect of IPL/SPL is a hub in the comparison of visual characteristics of the attended stimulus to category templates retrieved from memory, in this way acting as an important waystation in the attention-mediated classification and abstraction of visual input.

Interpretation of the cluster centered on the right insula is fundamentally challenging because of the breadth of functional roles associated with this area [59,60]. A compelling possibility; however, is that this reflects the role of insula and surrounding brain tissue in the "salience network", a set of brain areas implicated in the identification of homeostatic relevance and behavioral importance [61]. The salience network is thought to guide the flow of information through other brain networks, including prefrontal cortex, in the support of attentional processing and other cognitive operations [62]. Unlike IPL/SPL, category information in insula correlated discretely with ERP laterality in the latency of the N2pc, implicating the N2pc specifically in the propagation of information to this area. The emergence of attention-mediated category information in this cluster suggests a role for the insula and surrounding tissue in the processing of abstracted object information to guide the evaluation of behavioral relevance and the preparation of a cognitive response, possibly in the context of current participant state and task motivation.

The cluster centered on VMPFC is the largest of the three, has the strongest relationship with N2pc, and uniquely shows a link to both target-elicited and distractor-elicited N2pc, with target category information reducing in this area as a function of the mis-deployment of attention to the distractor. As with the insula, this cortical area has been associated with extensive functionality [63]. Prominent is the idea that it carries information about the abstracted economic utility of goods and actions—the "common currency" hypothesis [64,65]. This kind of signal is necessary for economically normative models of learning and decision-making to allow the utility of disparate goods and outcomes to be directly compared. However, our results show the emergence of attention-mediated category information in this area when categories do not explicitly differ in value. This is in line with the known involvement of this area in semantics [26], but is difficult to resolve with the notion that the core responsibility of this brain area is to represent value abstracted from context and stimuli. Building from other divergent findings in the literature, recent models of OFC/VMPFC function suggest that this brain region may rather be involved in the derivation of task state from disparate attributes of environmental objects and context [66,67] and in the establishment of cognitive maps of task space [68–70]. The current results are easily described in this theoretical context: strong attentional selection of a behaviorally-relevant visual object leads to the activation of a cognitive map in OFC/VMPFC that includes information about the target object category and its associated response. When attention is mis-deployed to the distractor, target information in this cognitive map is degraded.

As noted above, a motivation for Experiment 2 was to address the possibility that the relationship between N2pc and category information identified in results from Experiment 1 was driven by the encoding of visual features. In this context, one data pattern in Experiment 2 is notable for its absence. We do not see that the N2pc robustly predicts MRI-derived category information in visual cortex. This null finding is, of course, ambiguous, but suggests that our experimental design was successful in identifying information abstracted from the visual features that characterize the stimuli. That is, our stimulus set was visually heterogenous, with few low- or mid-level visual features that could be consistently used to infer category membership. As a result, categorization relied on a set of idiosyncratic diagnostic features for each image—for example, classification of a cupcake as food relied on resolution of a different set

of features than did classification of a courgette as food. Because attention acted on a varying set of relevant features for each image, attentional selection did not cause consistent benefit to a particular set of features, and therefore did not produce a consistent pattern of activity in visual cortex that could be leveraged by the classifier to improve performance. In this, results from Experiment 2 provide some reassurance that findings from unimodal EEG analysis in Experiment 1 are not driven entirely by the representation of visual features.

Importantly, this does not imply that the attention-mediated representations that we identify in the posterior parietal cortex, insula, and prefrontal cortex are necessarily amodal or instantiated independent of sensorimotor systems [71]. The relationship between N2pc and category information we identify is not a simple reflection of attentional effects on visual features but is likely to contain semantic features that are grounded in sensorimotor systems, including information about the specific motor response associated with each category [72]. The pattern we identify—that attentional effects on perceptual representations in the lateral occipital cortex predict the quality of perception-abstracted information elsewhere in the brain—is conceptually in close line with modern accounts of grounded semantic representation [73,74].

In addition to our identification of attention-mediated propagation of semantic information, results from Experiment 2 provided the opportunity to identify brain structures where N2pc amplitude predicted unimodal fMRI activation. This identifies a set of brain areas including left MFG (including the left frontal eye fields) [75], left IPL extending into TPJ, and bilateral dmSFG (Fig 7). All of these areas have documented roles in attentional control. As noted above, the FEF and IPL are core nodes in the frontoparietal network implementing strategic attentional control, and the broader MFG and lateral posterior parietal cortex have been implicated in the use of attention for semantic extraction. Similarly, dmSFG contains eye fields and plays a role in saccadic control. This suggests that activity in these areas may be involved in the establishment and instantiation of attentional control, such that their activation causes an increase in N2pc amplitude. This account is reinforced by the temporal pattern of the relationship between dmSFG and N2pc, and between IPL and N2pc, where laterality in the ERP preceding the N2pc window predicts activity in these areas (Fig 7B).

In conclusion, we track the impact of spatial attention on the spread of conceptual information in the human brain. Our results demonstrate the time-course of this propagation of information and identify three key brain structures carrying abstracted, attention-mediated information: the lateral posterior parietal cortex, the anterior insula, and the ventromedial prefrontal cortex. The emergence of the lateral posterior parietal cortex in this context is closely in line with prior research on spatial attention, but the identification of insula and VMPFC is unexpected and, as a result, constrains and guides functional interpretation of these structures in broader contexts. This brain network intervenes between perception and action, translating visual information to conceptual knowledge and action plans.

## Methods

### Human participants

All participants were recruited from the University of Birmingham community, provided written informed consent, reported normal or corrected-to-normal vision, and reported no history of neurological or psychiatric disorders. All demographics collected from experimental participants are reported and results can be expected to generalize to the European population from which the sample was taken. Experimental procedures were approved by the University of Birmingham STEM ethics committee (ERN_11-0429AP84) and the Centre for Human

Brain Health (CHBH) Health and Safety committee and adhered according to the principles expressed in the Declaration of Helsinki.

**Experiment 1.** Forty-nine participants were recruited. Two were excluded from analysis due to excessive eye movement artifacts in the EEG (rejection of >40% of trials), two were excluded as results showed no consistent evidence of a target-elicited N2pc (i.e., in the 1,140 iterations employed for classification, the lateral target-elicited signal showed contralateral positive polarity more often than negative polarity; per-participant $p < 0.05$, binomial test), and a single participant was excluded due to low accuracy in task performance (<70% accuracy; final sample of 44; age mean = 21.8, std = 2.7; 23 female; 3 left-handed). The experiment took approximately 2.5 h, including EEG preparation, task training, breaks, and debrief, and participants were compensated at £10/h.

**Experiment 2.** Thirty-nine participants were recruited. Two participants were excluded from analysis due to excessive eye movement artifacts in the EEG (rejection of >40% of trials), four participants were excluded due to incomplete data collection due to technical issues or equipment failure, and two participants were excluded due to low accuracy in task performance (<70% accuracy; final sample of 31; age mean = 23.2, std = 3.8; 19 female; 3 left-handed). The experiment took approximately 4 h, including EEG preparation, MRI screening, task training, breaks, and debrief, and participants were compensated at £15/h.

## Experimental task

Participants in both experiments performed a visual search task and a repetition detection task in interleaved blocks (4 blocks of 128 trials for each task with counterbalanced order). Completion of the visual search task took longer—approximately 75% of the experiment duration—and only this task is treated in the current paper.

In each experimental trial, participants were asked to report the semantic category of a target object (Fig 1A) that appeared in a cued location and to ignore a distractor object that appeared in an un-cued location (Fig 1B). Each trial began with a fixation display on white background (1,550–1850 ms, randomly selected from a uniform distribution) that contained four gray circles (E1: 5.7° visual angle; E2: 4.2°) directly above and below (E1: 7.4° visual angle; E2: 5.2°) a central fixation mark. The fixation mark was replaced by the letter "R" or "G" (1,350–1,650 ms), which indicated the red or green color of the circle identifying the target in the subsequent search array. In the search array, images of complete objects appeared within the two circles that had red or green color. The objects were tools, vehicles, animals, or foods, and were selected from an image set containing equal numbers of each category (E1: 160 objects in total; E2: 120 objects; Fig 1A) taken from the BOSS database [28] or open online image repositories. Image examples were selected from each category randomly without replacement until the set was exhausted and the process reset. The other two circles in the display contained unrecognizable morphed versions of the same image set [76] (max distortion 160, nsteps 4, iteration 9), selected in a similar manner. The search array sustained for 600 ms and the next trial began either immediately after response or after 1800 ms.

The experimental task was to identify, via right hand response, which object category appeared in the cued circle. Two categories were both associated with one response (of the index finger) and the other two were both associated with another response (of the middle finger), with the mapping of category to response counterbalanced across participants. This design feature was employed to make possible the verification of classification accuracy of MRI results in the ventral visual cortex and elsewhere when motor preparation and execution were equated across categories. In Experiment 1 response was made via a standard keyboard;

in Experiment 2 response was via an MRI-compatible button box. The response associated with the target object was always opposite to that associated with the distractor object.

The experiment began with verbal instructions and participants subsequently completed training until they achieved ~80% accuracy, which rarely took longer than a single block of trials. In Experiment 2 training took place outside of the scanner. Each experimental block was composed of 128 trials in random order, with each trial reflecting one combination of cue color, target and distractor category, and target and distractor position. In Experiment 1, participants were seated in a quiet room, and stimuli were presented via a 59.5 cm × 33.5 cm LCD monitor at 60 Hz and a distance of 60 cm. In Experiment 2, participants were supine, and stimuli were presented via a ProPiXX projector at 100 Hz viewed via a mirror mounted on the scanner head coil. The physical conditions between the two experiments were therefore dramatically different. As described above, stimuli were presented in a wider field of view in Experiment 1, and participants in Experiment 2 had fewer opportunities for breaks and were required to stay entirely still inside the scanner. These structural differences in experimental design and context are likely to be responsible for minor differences in EEG results across experiments. The experimental procedure relied on the PsychToolbox 3 toolbox [77] for Matlab.

## Data acquisition and preprocessing: Experiment 1

Electrophysiological data was acquired from sintered Ag/AgCl electrodes at 1 kHz using a Biosemi ActiveTwo amplifier. EEG was collected from 128 electrodes fitted in an elastic cap at equidistant encephalic sites, horizontal electrooculogram was collected from two electrodes located 1 cm lateral to the external canthi of the left and right eye, vertical electrooculogram was collected from two electrodes immediately above and below the right eye, and unused reference signals were collected from two electrodes placed over the left and right mastoid processes. EEG was resampled offline to 200 Hz, digitally filtered with symmetric Hamming windowed finite-impulse response kernels (high-pass at 0.05 Hz, −6 dB at 0.025 Hz; low-pass at 45 Hz, −6 dB at 50.6 Hz), referenced to the average of encephalic channels, and baselined on the 200 ms interval preceding stimulus onset. Noisy channels were visually identified and interpolated using spherical spline interpolation. Independent component analysis (ICA) [78] of combined EEG and EOG data was used to identify data variance resulting from eye movements. Trials with eye movements in the 500 ms interval following stimulus onset were removed from analysis and the ICA components reflecting eye artifacts were subsequently removed from the data. After removal of noisy and incorrect trials 429 trials were left per participant (mean; 48 SD).

## Data acquisition and preprocessing: Experiment 2

**MRI.** Whole-brain scanning employed a 3T Siemens Prisma MRI scanner and 64-channel head-coil. Functional data was acquired using an echo planar imaging (EPI) sequence with 57 axial slices of 84 × 84 voxels, field-of-view (FOV) 210 mm × 210 mm, slice thickness 2.5 mm, slice gap 0 mm, repetition time (TR) 1.5 s, echo time (TE) 35 ms, flip angle (FA) 71°, multi-band acceleration factor 3. Structural data was acquired using a T1-weighted MPRAGE sequence with 208 axial slices of 257 × 257 voxels, FOV 256 mm × 256 mm, slice thickness 1 mm, slice gap 0 mm, TR 2 s, TE 2.03 ms, FA 8°. For correction of other images, a field map was acquired using a double-echo sequence with 36 axial slices of 64 × 64 voxels, FOV 192 mm × 192 mm, slice thickness 3.75 mm, slice gap 0 mm, TR 400 ms, $TE_1$ 4.92 ms, $TE_2$ 7.38 ms, FA 45°. Four-lead vectorcardiogram (VCG) was recorded in the scanner from electrodes in a standard chest montage. VCG was recorded in synchronization with the MRI clock.

MRI images were corrected for head movement, field map inhomogeneity, and slice-acquisition delay (using the middle slice as reference) before being normalized to Montreal Neurological Institute space, interpolated to 2 mm isotropic voxel size, and spatially smoothed using an isotropic Gaussian kernel (6 mm full-width half maximum). General linear model (GLM) analysis of the fMRI time-series data relied on the GLMsingle toolbox [39]. GLMsingle identifies hemodynamic kernel functions, derives nuisance regressors, and chooses an optimal ridge regularization shrinkage fraction for each voxel to produce an estimate of single-trial voxel-wise beta values. To align EPI volumes with the onset of visual stimulation, the EPI time series was up-sampled to 6.7 Hz (150 ms per image) prior to linear modeling using shape-preserving piecewise cubic interpolation (as implemented in the tseriesinterp.m function included in knkutils toolbox; https://github.com/cvnlab/knkutils). Analysis was constrained to voxels outside of the brainstem, the ventricles, and the cerebellum.

**EEG.** Electrophysiological data was acquired in the scanner from sintered Ag/AgCl electrodes using a Brain Products BrainAmp MR amplifier. EEG was collected from 63 electrodes fitted in a BrainCap MRI-compatible elastic cap at extended 10/20 encephalic sites and single-lead electrocardiogram (ECG) was recorded from an electrode located on the back 10 cm caudal to the top of the shoulder and 5 cm to the left of the spine. EEG was recorded at a 5 kHz sample rate using a Brain Products SyncBox, which phase locks the EEG sample rate with the 10 MHz internal clock of the MRI scanner. Preprocessing began with correction for MR gradient artifacts using BrainVision Analyzer 2 (v2.2). Gradient triggers sent by the scanner were used as time markers for artifact trains and continuous artifacts were corrected using template subtraction based on a sliding average of 21 TR intervals (with baseline correction). The EEG was subsequently downsampled to 250 Hz and corrected for cardioballistic (CB) artifacts. The CB correction algorithm semi-automatically detects peaks from heartbeat signals. In the 124 experimental blocks analyzed (4 for each of 31 participants), the signal quality and availability of heartbeat signals varied: VCG was used for correction in 52 blocks, ECG was used in 48 blocks, and the remaining 24 blocks relied on a heart-beat ICA component derived from the EEG. A running average of 21 beats was used to generate a template and the artifact time delay was automatically estimated using a 40 s interval. Subsequent digital filtering, baselining, noisy channel interpolation, ICA, and trial rejection parameters were as described for Experiment 1. The average number of interpolated channels was 0.29 and the number of independent components removed was 2.0. The mean number of correct trial epochs containing excessive noise or eye movements was 27 (SD = 26), and the mean number of incorrect trials was 52 trials (SD = 36). These excessively noisy trials were excluded from analysis, as were incorrect trials, leaving 433 trials per participant (mean; SD = 41).

## Quantification and statistical analyses

**Measuring category information in EEG.** In both experiments, machine-learning classification of multichannel EEG data relied on LDA and cross-fold validation. As noted above, EEG classification relied on temporal searchlights defined across a window of ~ 85 ms (17 samples in Experiment 1; 21 samples in Experiment 2). Trials were not averaged prior to classification. For classification of both EEG and MRI, each data feature in the training set (i.e., voltage at one EEG channel or activation of one voxel) was $z$-scored across observations. The mean and standard deviation values derived from normalization of the training set were retained and used in normalization of the test set. Classification of EEG and MRI data relied on the COSMOMVPA toolbox for Matlab [79].

In Experiment 1, we adopted a validation scheme in the classification of EEG that involved partitioning of trials into 20 chunks, with 3 and 17 chunks exhaustively allocated into training

sets and test sets over 1,140 iterations. In Experiment 2, trials were partitioned into 10 chunks, with 2 and 8 chunks exhaustively allocated into test and training sets over 45 iterations. In both experiments, the classification results reflect mean accuracy over iterations and participants (Fig 2C and 2F). Statistical significance reflects the analysis of between-participant variance using cluster-based correction over time with cluster-defining threshold at $p < 0.05$ and cluster significance at $p < 0.05$ [80]. Conditional differences in classification latencies are directly interpretable, but the absolute values should be interpreted with care as the filtering of EEG and use of temporal searchlights meant that the classifier had access to EEG data recorded up to ~50 ms later than the nominal latency.

**Coupling EEG indices of attentional selection to variance in EEG-derived category information.** As described in the results section, we coupled N2pc amplitude to EEG-derived target category information using an iterative resampling approach. This garnered 1,140 value pairs containing mean N2pc amplitude and mean classification accuracy for each participant. The relationship between these variables was analyzed using mixed linear modeling as implemented in the fitlme.m function for Matlab (statistics toolbox R2022a). In mixed linear modeling, restricted maximum likelihood was used for variance estimation, AIC was used for model comparison, and ANOVA derivations employed Satterthwaite approximations of degrees of freedom.

**Measuring category information in unimodal fMRI.** As in the analysis of EEG, machine-learning classification of MRI results labeled the target or distractor category in individual trials using LDA and cross-fold validation. Results for target-lateral and distractor-lateral conditions were not separated in the classification of MRI data. Searchlight spheres were constrained when bound by the edge of brain space and the mean sphere volume was 111.8 voxels. Classification was completed using two validation schemes. In the first, 4-fold cross-validation was employed to generate the results illustrated in Fig 5. Subsequently, 1,140-fold cross-validation was used to identify covariance in MRI classification and N2pc. Mean MRI classification results for 4-fold and 1,140-fold classification were very similar.

Statistical analysis of classification results employed $t$-tests of searchlight classification accuracy against a null of chance performance. Results were cluster-corrected for multiple comparisons with a cluster-defining threshold of $p < 0.001$ and cluster significance at $p < 0.05$ [80,81]. Statistical contrasts of target versus distractor classification accuracy relied on similar parameters, but with $t$-tests contrasting results from the two analyses.

**Coupling variance in EEG indices of attentional selection to variance in fMRI-derived semantic category information.** Identification of the relationship between N2pc amplitude and fMRI-derived target category information followed the same analytic schema as in analysis of EEG from Experiment 1, with the added dimension of searchlight location. Statistical analysis of results from this analysis employed one-tailed $t$-tests of correlation values against a null hypothesis of zero. Results were cluster-corrected for multiple comparisons with a cluster-defining threshold of $p < 0.001$ and cluster significance at $p < 0.05$.

**Coupling variance in EEG indices of attentional selection to variance in fMRI-derived semantic category information: time-course.** In ROI analysis, regions were defined based on results from searchlight analysis illustrated in Fig 6. We applied our 1,140-fold classification technique to the voxel pattern extracted from each of these ROIs, generating 1,140 value pairs containing mean N2pc and mean classification accuracy for each ROI. Fisher-transformed Pearson correlations were calculated for each ROI and results for each ROI were statistically assessed using a permutation test with 50k iterations. Statistical analysis of time-course results within each cluster relied on temporal cluster correction with a cluster-defining threshold of $p < 0.05$ and cluster significance at $p < 0.05$.

**Coupling variance in EEG indices of attentional selection to variance in fMRI voxel activity.** Identification of the relationship between N2pc amplitude and voxel activity

followed much the same schema as described immediately above, but assessing voxel activity rather than searchlight classification accuracy. In ROI analysis, regions were defined based on results illustrated in Fig 7. Statistical analysis of these results relied on the same permutation tests described above, and statistical analysis of time-course results relied on cluster correction with the same parameters described above.

## Acknowledgments

Our thanks to Vinura Munasinghe, Holly Ahmed, and Tia Cainer for support with EEG data collection; to Steve Mayhew for technical expertise on the recording of concurrent EEG/MRI; to Alicia Rybicki, Alan Kingstone, Marius Peelen, and Wieske van Zoest for discussion; and to the University of Birmingham's BlueBEAR high-performance computing service (www.birmingham.ac.uk/bear) for computational resources.

## Author contributions

**Conceptualization:** David Acunzo, Clayton Hickey.

**Formal analysis:** David Acunzo, Clayton Hickey.

**Funding acquisition:** Clayton Hickey.

**Investigation:** David Acunzo, Damiano Grignolio.

**Methodology:** David Acunzo, Damiano Grignolio, Clayton Hickey.

**Software:** David Acunzo, Clayton Hickey.

**Supervision:** Clayton Hickey.

**Visualization:** David Acunzo, Clayton Hickey.

**Writing – original draft:** Clayton Hickey.

**Writing – review & editing:** David Acunzo, Damiano Grignolio.

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
