## [Editor Report · Decision Letter 0]

8 Aug 2024

Dear Clayton, 

Thank you for submitting your appeal for your manuscript entitled "Neural mechanisms for the attentional propagation of conceptual information in the human brain" for consideration as a Research Article by PLOS Biology. Apologies that it has taken me longer than it should to get back to you due to the busy summer period.

Your manuscript has now been evaluated again by the PLOS Biology editorial staff and also by an academic editor with relevant expertise and I am writing to let you know that we would like to send your submission out for external peer review.

Once your full submission is complete, your paper will undergo a series of checks in preparation for peer review. After your manuscript has passed the checks it will be sent out for review. To provide the metadata for your submission, please Login to Editorial Manager (https://www.editorialmanager.com/pbiology) within two working days, i.e. by Aug 10 2024 11:59PM.

Kind regards,

Christian

Christian Schnell, PhD

Senior Editor

PLOS Biology

cschnell@plos.org

---

## [Decision Letter · Decision Letter 1]

4 Oct 2024

Dear Clayton,

Thank you for your patience while your manuscript "Neural mechanisms for the attentional propagation of conceptual information in the human brain" was peer-reviewed at PLOS Biology. It has now been evaluated by the PLOS Biology editors, an Academic Editor with relevant expertise, and by several independent reviewers. 

In light of the reviews, which you will find at the end of this email, we would like to invite you to revise the work to thoroughly address the reviewers' reports.

As you will see below, the reviewers overall think that the study is well executed and provides important insights. While Reviewer 1 and Reviewer 3 ask for a few clarifications and additional analyses, Reviewer 2 has several concerns about the experimental approach and the theoretical framing, that need to be addressed.

Given the extent of revision needed, we cannot make a decision about publication until we have seen the revised manuscript and your response to the reviewers' comments. Your revised manuscript is likely to be sent for further evaluation by all or a subset of the reviewers.

**IMPORTANT - SUBMITTING YOUR REVISION**

*Re-submission Checklist*

*Published Peer Review*

*PLOS Data Policy*

*Blot and Gel Data Policy*

Sincerely,

Christian

Christian Schnell, PhD

Senior Editor

PLOS Biology

cschnell@plos.org

REVIEWS:

Reviewer #1: The authors present interesting data from an innovative study that relates variation of the N2pc component to representation of categorical/conceptual information. Across two experiments, elegant multivariate analyses are employed to reveal how attention influences the propagation of conceptual information in the brain. Experiment 1 is a pure EEG experiment, whereas Experiment 2 extends a similar logic to an EEG-fMRI experiment. The results are rich and interesting, overall shedding light on how attention guides the abstraction of information in the human brain.

I enjoyed reading this manuscript. As best I can tell, the methods are innovative and rigorous (one small comments on thresholding below), and the results are systematic and interesting. I think the manuscript can make an important contribution both on the content level, as well as methodologically. I do have a couple of comments, however, which I would ask the authors to address.

1) The authors acknowledge that variation in the N2pc could reflect various processes (including misallocation of attention), but the question of the source of this variation is not discussed (or explored) in great detail. Specifically, I was wondering in how far results may be in particular driven by full misallocation in the sense of having attended and responded to the wrong item (of course there are other ways of making a mistake, but since the responses expected for the target and distractor were opposite, it seems likely that wrong responses often simply reflect responding to the wrong stimulus), which would likely have a strong influence on the results that would go beyond "increasingly robust attentional selection of the target", which implies something more gradual. In this context, I am not fully sure whether incorrect trials were excluded (sorry, if I missed this information). So, just to be sure, were they?

2) Fig. 2: I think the EEG data recorded inside the scanner looks really good; to further illustrate this, I think it would be good to add a topo plot to panel D, just like the one in panel A. Then, you could also more clearly indicate that the sensor plots on the left and on the right rely on slightly different channels (2 sets of three electrodes for the EEG-only experiment, and only 1 electrode on each side for the EEG-fMRI dataset, if I understood correctly).

3) I was not sure why the current set-up yielded a distractor N2pc, rather than a distractor positivity. It is quite possible that I am misunderstanding something, and/or it may relate back to my first question, but it may be useful to clarify.

4) It would also be good to clarify a bit how the MR and EEG data were synchronized during acquisition. It is stated that the EEG (and VCG) data was acquired at MR clock speed (5 kHz), but I believe that the MR clock is much faster. Without any further synchronization (see below), also the choice of number of slices vis-à-vis the TR could have been a problem (since a slice duration results that is not divisible by the EEG sampling rate, which is not recommended, I think). However, you also mention that the clock was synchronized, which may refer to the use of an additional Brainproducts sync-box. I think it would be good to check and potentially explain this a bit more to avoid confusion. As I said before, the EEG data from inside the scanner looks good, implying that it was set up correctly, so that I assume that this is just about clarification.

5) The thresholding approach is inconsistent, with cluster-forming thresholds being either p<.001 or p<.05, which is very different. The latter is also explicitly discouraged, if I am not mistaken: https://pubmed.ncbi.nlm.nih.gov/27357684/

Reviewer #2: Summary:

This paper presents two experiments investigating the relationship between early attentional deployment and the categorization of objects that were spatially attended vs. not attended in a search-like display. Using a combination of ERP markers and fMRI data, this study presents a novel and interesting data set that in a first experiment establishes a relation between early attentional selection (marked by the N2pc component of the ERP) and object categorization using EEG classification data; in a second experiment, the N2pc is further associated with activity patterns in various brain regions. Overall, this study undoubtedly presents a rich and interesting new data set. At the same time, it's a bit unclear what the major theoretical contributions are and what some of the N2pc-to-brain regions links mean. I also had several clarification questions with regards to the analyses which were often described rather briefly. Below are my detailed comments and questions.

Also I want to note that the manuscript file was named _R1, so I assume I was brought in as a reviewer in the second round of revisions. However, I did not see the revision or response letter to the first round of revisions, so treating this like a new manuscript.

Introduction/Motivation

1) I am not sure I agree that the "emergent [attention] model suggests that strategic selection begins with the derivation of schemas and templates from conceptual knowledge" (pg. 3). There is quite a good amount of attention research suggesting / arguing that low-level features and locations are what guides attention and selection occurs based on these lower-level representations that may feed into higher-level representations, but nonetheless play a critical role in how attention is allocated (For example see earlier TICS paper by Chapman et al. 2024 on these topics).

2) A related point: "We presume that information selected through the deployment attention activates or constructs neural representations of concepts and categories" (pg., 3/4, abstract etc) — who is "we" in this context? Is this a reference to the field of attention researchers, or the present authors? Again, I wasn't sure what the evidence is for this statement and motivation here— it's not super clear that this is a common theory of what attention does.

Methods/Results:

3) Why were the scrambled objects presented? I assume to match stimulus presentation across hemispheres, but as the authors allude to, the morphed vs. real object images are not perfectly matched. Also it says the objects were chosen to be "visually different". Was this verified using a model, e.g., CNNs, or was this decided visually by the authors?

4) Linking of EEG to fMRI data: I may have missed this, but how do the authors control for multiple comparisons given many brain sites they are testing for correlating with the EEG data?

5) Do I understand correctly that participants were asked to do a 4-way category discrimination, yet only used two response buttons such that 2 categories were mapped onto the same button? That seems a bit odd if the question is about categorization, as really what participants needed to do it group two categories to respond correctly?

6) It says trials were not averaged prior to classification — but the "chunking" refers to averaging across small portions of trials, is that correct? I just want to make sure it's clear whether the classifier uses small subsets of averaged EEG data (ERPs) or single-trials. 

7) For the classification, the data is split into target-lateral and distractor-lateral trials, following the same logic of the N2pc component. Thus , if I understood correctly, training and testing data sets are different, which makes it somewhat difficult to interpret the classification accuracy of the EEG data (e.g., Fig. 2C). It could be that the difference is driven by differences in the training set due to various reasons (e.g., lower quality of the data overall, or some other variable that is different than the one of interest here). Ideally, the training set would be "neutral" condition, and then testing would occur on attended and unattended objects. I realize that's not possible in the current task design, but have the authors tried to train on all the data and then test either on target- or distractor-lateralized activity? 

8) Did the FMRI classification use a similar approach to train and test for lateral-target and lateral-distractor? It was not exactly clear to me how target vs distractor activity was extracted in this case, and whether laterality should matter for the different brain regions.

9) The reported links between the N2pc and various brain regions is interesting, and the authors are doing a good job trying to walk through what these links could mean. Yet, I couldn't fully follow or understand the theoretical implications of the three main brain areas found: lateral PPC, insular, and VMPC. As the authors say themselves, these areas have been implicated in various functions and tasks, and so I was not convinced that this allows drawing the conclusion that this shows that spatial attention is directly linked to object categorization. Maybe the authors can be more careful here.

Minor:

10)When reading the manuscript and first looking at the ERP data, I immediately thought about the fact that the N2pc effect appears sustained (or N2pc followed by a CDA-like component); the authors later on report analyses across the longer time interval, but it might be nice to announce this earlier on.

Reviewer #3: The study aims to identify the temporal and anatomic characteristics of the propagation of attention-mediated conceptual information in the human brain. To address this question, the authors carry out two experiments, one unimodal EEG experiment, and a second experiment with concurrently-recorded EEG and fMRI, with a similar task performed by the participants in both experiments. The recorded data is valuable and provides the advantage of looking at both the temporal and spatial dimensions of attentional effects on semantic content formation in the brain. The authors perform several analysis steps, looking at the N2pc signal in EEG data, target category classification in fMRI data, and the relationship between these two signatures of attention using the concurrently-recorded EEG and fMRI data. They show the relationship between the target-elicited N2pc in EEG and the target classification accuracy in fMRI data. They further identify brain regions in which target category information is predicted by target-elicited N2pc amplitude.

One strength of the study is the use of both EEG and fMRI data, providing the opportunity to look at the temporal dynamics in addition to providing a good spatial resolution.

Another strength is that they show the relationship between the target-elicited N2pc and fMRI data both at the univariate level (voxel activity) and at the multivariate level (target classification).

Identifying the brain regions that are suggested to play a role in the emergence of conceptual information is an important result.

The manuscript would benefit from addressing these points:

1- Considering the relationship between N2pc amplitude and fMRI target category classification accuracy, the authors draw a causal conclusion from a positive correlation (e.g. in page 9 in the last paragraph and page 12). While they provide the lack of a significant relationship between distractor-elicited N2pc and target information as justification, this is still not enough justification for a causal relationship. The authors should either tone down about the causal relationship throughout the manuscript, or provide further stronger justification.

2- Page 14, section on "Coupling variance in lateral EEG tovariance in fMRI-derived category information":

Some discussion is needed on why there is difference between the three ROIs in the latency of significant correlation between lateral EEG and fMRI category information. The section talks about the results of the correlation and concludes in a sentence that these results suggest a close link between selective processes in N2pc and the propagation of target-category information to these areas. However, the results in Figure 6 show that the relationship between EEG laterality and target category information emerges at different times with different durations in the three ROIs. These differences should be addressed more explicitly. It should also be discussed whether this difference affects the conclusion.

3- The authors should address the difference in the ERP and classification results of the two EEG experiments (Fig. 2). This includes the differences observed in the lateral occipital voltage (Fig. 2a and 2d) and contralateral-ipsilateral voltage (Fig. 2b and 2e), as well as in EEG classification accuracy (Fig. 2c and 2f).

4- -Details about stats are missing in some parts (e.g. p. 16, second paragraph)

5- Figure 2 caption b and e: should be "contralateral minus ipsilateral" instead of "ipsilateral minus contralateral"

6- References 27 and 48 are the same, also references 15 and 26.

7- Reference 37 is under review. Maybe provide the information of the bioRXiv version?

---

## [Decision Letter · Decision Letter 2]

19 Dec 2024

Dear Clayton,

Thank you for your patience while we considered your revised manuscript "Neural mechanisms for the attentional propagation of conceptual information in the human brain" for publication as a Research Article at PLOS Biology. This revised version of your manuscript has been evaluated by the PLOS Biology editors, the Academic Editor and the original reviewers.

Based on the reviews and on our Academic Editor's assessment of your revision, we are likely to accept this manuscript for publication, provided you satisfactorily address the remaining points raised by the reviewers. Please also make sure to address the following data and other policy-related requests:

* We would like to suggest a different title to improve its accessibility for our broad audience: "Neural mechanisms for the attention-mediated propagation of conceptual information in the human brain"

* Please add the links to the funding agencies in the Financial Disclosure statement in the manuscript details.

* Please include information in the Methods section whether the study has been conducted according to the principles expressed in the Declaration of Helsinki.

* DATA POLICY:

CODE POLICY

We expect to receive your revised manuscript within four weeks. 

*Published Peer Review History*

*Press*

Sincerely,

Christian

Christian Schnell, PhD

Senior Editor

cschnell@plos.org

PLOS Biology

Reviewer remarks:

Reviewer #1: The authors have undertaken a thorough and thoughtful revision. All of my points were addressed. 

That being said, personally, I would add the topo plot of the N2pc difference from within the scanner to the manuscript (#2 of my review), and provide a short version of the explanation for the more complex voltage distribution as a footnote. I think this is useful information, and also illustrates some of the complexity of running MR-EEG experiments. In my mind, it also does not undermine the inferences of the paper at all. In case you end up adding it, make sure to check again whether it is a left-minus-right or right-minus-left difference. This was inconsistent across the caption you provided and the text. 

Reviewer #2: The authors have addressed my questions in their response letter.

---

## [Editor Report · Decision Letter 3]

14 Jan 2025

Dear Clayton,

Thank you for the submission of your revised Research Article "Neural mechanisms for the attention-mediated propagation of conceptual information in the human brain" for publication in PLOS Biology. On behalf of my colleagues and the Academic Editor, Ed Vogel, I am pleased to say that we can in principle accept your manuscript for publication, provided you address any remaining formatting and reporting issues. These will be detailed in an email you should receive within 2-3 business days from our colleagues in the journal operations team; no action is required from you until then. Please note that we will not be able to formally accept your manuscript and schedule it for publication until you have completed any requested changes.

While you attend to those changes, please also ensure the dataset at edata.bham.ac.uk/id/user/595 becomes available, so we can check it before publication of your paper.

PRESS

Sincerely, 

Christian

Christian Schnell, PhD

Senior Editor

PLOS Biology

cschnell@plos.org